# Individual and public health consequences associated with a missed diagnosis of pulmonary tuberculosis in the emergency department: A retrospective cohort study

Courtney Heffernan[1]*, Catherine Paulsen[1], Leyla Asadi[1], Mary-Lou Egedahl[1], Brian H. Rowe[2,3], James Barrie[4], Richard Long[1,3]

1 Tuberculosis Program Evaluation and Research Unit, Department of Medicine, University of Alberta, Edmonton, Alberta, Canada, 2 Department of Emergency Medicine, University of Alberta, Edmonton, Alberta, Canada, 3 School of Public Health, University of Alberta, Edmonton, Alberta, Canada, 4 Department of Radiology and Diagnostic Imaging, University of Alberta, Edmonton, Alberta, Canada

* cheffern@ualberta.ca

**Data Availability Statement:** The demographic and clinical data underlying the results presented in this study are available from: https://figshare.com/

## Abstract

### Objectives

To determine: i) the emergency department (ED) utilization history of pulmonary tuberculosis (PTB) patients, and ii) the potential individual and public health consequences of a missed diagnosis of PTB in this setting.

### Design

Retrospective observational cohort study.

### Participants

Patients with PTB aged >16 years diagnosed between April 1, 2010 and December 31, 2016 in the Province of Alberta, Canada.

### Methods

We identified valid new cases of PTB from a provincial registry and linked them to ED attendees in administrative databases. Visits are considered 'PTB', pulmonary 'other', and non-pulmonary based on the most responsible discharge diagnosis. Individual consequences of a missed diagnosis included health system delay and PTB-related death; public health consequences included nosocomial ED exposure time and secondary cases.

### Results

Of 711 PTB patients, 378 (53%) made 845 ED visits in the six months immediately preceding the date of diagnosis. The most responsible ED discharge diagnosis was PTB in 92 (10.9%), pulmonary 'other' in 273 (32%) and non-pulmonary in 480 (56.8%). ED attendees had a median (IQR) health system delay of 27 (7,180) days and, compared to non-ED attendees were more likely to die a TB-related death 5.9% vs 1.2%, p = 0.001. Emergency

articles/dataset/Supplemental_Data_Emergency_
Department_Use_by_PTB_Patients_in_Alberta_
2010-2016/14077310.

**Funding:** This work received financial support in
the following forms: An Alberta Lung Association/
Respiratory Health Strategic Clinical Health
Network Studentship (SFR1218), and University of
Alberta Pulmonary Research Group award
provided financial support to CH for her studies, of
which this work represents one part. BR's research
is supported by a Scientific Director Grant (SOP
168483) from the Canadian Institutes of Health
Research (CIHR; Ottawa, ON).

**Competing interests:** The authors have declared
that no competing interests exist.

attendees generated 3812 hours of ED nosocomial exposure time, and 31 secondary cases
(60.8% of all secondary cases reported). *Mycobacterium tuberculosis* isolates from ED-
attendees were more likely than non-attendees to be clustered–i.e., have an identical DNA
fingerprint with another isolate (27% vs. 21%, p = 0.037).

## Conclusions

ED utilization by PTB patients, and related consequences, are substantial. EDs are a poten-
tial resource for earlier PTB diagnosis.

## Introduction

Canadians frequently seek care in emergency departments (EDs). In 2016, a Commonwealth
Fund survey found Canadians were more likely to seek care in an ED than citizens of ten com-
parator countries [1]. Emergency department use by patients with tuberculosis (TB) in high-
income, low-incidence countries, like Canada, is well- documented [2–9]. Relatively rare con-
ditions like TB are challenging to diagnose when patients present with non-specific complaints
and/or in high throughput settings; missed diagnoses generate individual consequences, or, as
with a missed diagnosis of pulmonary TB (PTB), the communicable form of TB, both individ-
ual and public health consequences.

   Patients with PTB can have non-specific presenting complaints, and ED physicians without
pulmonary or infectious disease training may be unfamiliar with the combination of clinical,
epidemiologic, and radiographic features, which, together, suggest the diagnosis. Notwith-
standing these considerations, the setting is ideal and critical for making a timely diagnosis of
PTB. Timelier diagnoses, in turn, improve individual patient outcomes and, by earlier inter-
ruption of transmission, contribute to TB elimination. The ED setting has simple and
advanced diagnostic capabilities, which may extend to invasive diagnostic procedures like
bronchoscopy, thoracentesis, induced sputum, and specialist consultations.

   Previous studies have shown that ED use by patients with TB (all forms) was substantial;
use, however, likely varies by hospital, location, country, and other factors [2–9]. In the present
study, we sought to confirm the extent of use over a large jurisdiction (>100 EDs) in adult
patients with PTB. We further sought to measure the potential individual and public health
consequences of a missed PTB diagnoses in these ED patients. We hypothesize that province-
wide ED use by undiagnosed patients with PTB remains substantial, and that the ED is an
untapped case finding setting.

## Materials and methods

### Setting

We undertook this study in the Province of Alberta, one of four major immigrant receiving
provinces in Canada, with a population of 4,252,900 in 2016 (Statistics Canada) and a crude
TB incidence rate of 5.5/100,000 population that same year [10].

### Design

We undertook a retrospective observational cohort study. The cohort includes consecutively
diagnosed culture-positive patients with PTB aged >16 years, between April 1, 2010 and
December 31, 2016 through a validated provincial registry.

## Cohort

The study cohort included ED and non-ED attendees. Patients who registered in an ED at least once within six months of their confirmatory PTB diagnosis were designated as ED-attendees. The PTB diagnosis date was the start date of treatment, or the date of death in those who died without treatment. The lower limit for 'adult' emergency medicine in Alberta is 17 years. We identified cases from the Integrated Public Health Information System (iPHIS), which houses the Provincial TB registry. We supplemented clinical and demographic information with mycobacteriologic data from Alberta Precision Laboratory, where all provincial mycobacteriology is performed.

## Data linkage

We confirmed attendance at one or more of the EDs in the province by cross-referencing each case with the Alberta Health Services Data Integration Measurement and Reporting (DIMR) system using the personal health number (PHN). Alberta provides universal healthcare insurance and no co-payments are required to receive care.

## Data collection

Demographics included age at diagnosis, sex, population group (Canadian-born, foreign-born), and place of residence (urban vs. rural). Canadian-born cases included Indigenous peoples, whose Indigenous identity is self-reported at diagnosis and affirmed in the TB registry (e.g., First Nations, Métis and Inuit), and non-Indigenous Canadians. Cases with an urban residence reported a postal code at diagnosis in either one of Alberta's two major metropolitan areas–Edmonton, and Calgary. Clinical information included disease type (new active vs. relapse/retreatment), disease site (pulmonary alone vs. pulmonary plus extra-pulmonary), HIV status (positive vs. negative), presence or absence of diabetes, and the presence or absence of social risk factors. Social risk factors included any one of homelessness within the previous 12- months, a substance misuse disorder, and/or incarceration at the time of diagnosis. Mycobacteriologic information included smear-status (positive vs. negative) and drug susceptibility test results (resistant [any first-line drug resistance] vs. susceptible). Outcomes included treatment outcome (survived vs. died before completion of treatment), and transmission history—whether the patient did or did not generate any secondary cases.

To determine whether ED attendees and non-attendees generated any secondary cases, we used conventional and molecular epidemiologic data. The latter included the DNA fingerprints of initial isolates from all members of the cohort, and all other culture-positive cases diagnosed in the province between October 1, 2009 and December 31, 2018. This transmission window extends from six-months before the diagnosis of the first cohort case to 24-months after the diagnosis of the last one [11–13]. We performed secondary case analyses according to an established protocol [14, 15]. For patients diagnosed before July 1, 2010 we used restriction fragment-length polymorphism (RFLP) supplemented by spoligotyping in isolates with < 6 copies of IS*6110* as our fingerprinting method. For patients diagnosed on or after June 1, 2014 we used mycobacterium interspersed repetitive units (MIRU)–variable number tandem repeats (VNTR). To further estimate transmission risks among ED attendees and non-attendees, we compared the extent of genotypic clustering within each group. We defined a cluster as two or more cases with genotypically identical isolates; we considered members of the same cluster to belong to a TB transmission chain.

Using additional administrative data, we described the ED attendees according to the timing of their visits relative to their date of diagnosis, the duration of their ED visit, and the most responsible discharge diagnosis classified as: PTB, pulmonary 'other' (a pulmonary complaint

or problem other than PTB), or non-pulmonary (a non-pulmonary complaint or problem). We calculated a nosocomial exposure time among attendees with either pulmonary 'other' and non-pulmonary diagnoses at discharge as hours from registration or triage, whichever occurred first, until discharge.

Finally, we applied the three diagnostic triggers (symptoms, epidemiological risk, and typical chest radiograph with or without cavitation) to determine the actual and potential diagnostic yield for smear-positive and smear-negative PTB among ED attendees whose last visit occurred within 30 days of their definitive diagnosis. We define patients as 'symptomatic' if their discharge diagnosis was either PTB or pulmonary 'other'. A university-based chest radiologist blinded to the diagnostic group of the patient re-read chest radiographs, describing them as typical for adult-type PTB with or without cavitation, or atypical [14].

## Statistical analysis

We compared demographic, clinical, mycobacteriologic, and outcome information in ED-attendees vs. non-attendees. We report continuous data as means and standard deviations (SD) or median and interquartile range (IQR), as appropriate. We report dichotomous data as counts and percentages. We used student t-test and Pearson's chi square to test for significant associations between patients' clinical and demographic characteristics and use of an ED. We performed univariate tests to explore differences between ED attendees with multiple vs. single ED visits, and Pearson's chi square to test differences in clustering. We performed statistical analyses using STATA 12.0 (STATA Corp., College Station, Texas, USA).

## Ethics approval

The Health Research Ethics Board (HREB) at University of Alberta approved this study (approval number: Pro00076709). Alberta Health Services (AHS) provided administrative and operational approvals. We obtained a data-sharing agreement from AHS for the abstraction of data from the administrative data holding—DIMR. Given the retrospective design of this study, treating physicians and patients were unaware and not influenced by this study at the time of encounter.

## Results

During the study period, 711 patients were diagnosed with PTB in Alberta of whom 378 (53.2%) made one or more ED visits in the six months preceding their PTB diagnosis (ED attendees). ED attendees made 845 total visits; 184 (48.6%) made a single visit, while 194 (51.4%) made multiple visits. Just over half (431, or 51%) of all ED visits were made to five EDs–two in each of the major metropolitan areas, and one in a regional city.

As patients approached their date of diagnosis, they were increasingly more likely to make an ED visit (Fig 1). The most responsible discharge diagnosis was PTB in 92 (10.9%) visits, pulmonary 'other' in 273 (32%) visits, and non-pulmonary in 480 (56.8%) of visits. Visits in months 1–3 (prior to diagnosis) were more likely than those in months 4–6 (prior to diagnosis) to have a PTB or pulmonary 'other' discharge diagnosis (38.6% vs. 4.4%, p <0.0001). The median time (IQR) to diagnosis (treatment) was 2 (IQR: 0, 5) days after visits with a discharge diagnosis of PTB, 20 (IQR: 0, 52) days for pulmonary 'other' and 41 (0, 103) days for non-pulmonary diagnoses. Of those ED visits that ended with a pulmonary 'other' discharge diagnosis (n = 273), the leading diagnoses were pneumonia (39%), cough or hemoptysis not yet diagnosed (NYD) (15%), other respiratory symptoms (chest pain, dyspnea) NYD (14%), pleural effusion (6%), and abnormal chest radiograph NYD (4%).

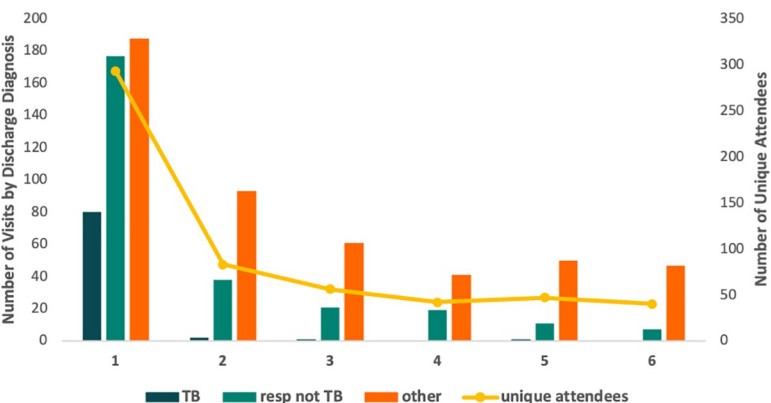

**Fig 1. Number of visits and number of unique attendees in the months immediately preceding the start date of treatment grouped according to ED discharge diagnosis.**

As shown in Table 1, ED-attendees were more likely than non-attendees to be Canadian-born (24.8% vs. 10.8%), have a rural place of residence (32.6% vs. 19.9%), have both PTB and extra-pulmonary TB (21.2% vs. 11.8%), be smear-positive (58.9% vs. 39.3%), to have a social risk factor (11.1% vs. 3.9%), and to die before treatment completion (6.3% vs. 1.2%). Among those whose death was TB-related (i.e. TB was either the primary or contributory cause of death) 22 (5.9%) of ED-attendees and 4 (1.2%) non-attendees died, respectively (p = 0.001). The 22 ED-attendees who went on to suffer a TB-related death had made 69 total visits, and none received a discharge diagnosis of PTB at those visits. ED-attendees and non-attendees did not differ by age, sex, presence of drug resistance, HIV status, presence of diabetes, or proportion of cases with one or more secondary cases.

Among ED-attendees, 22 generated one or more secondary cases. In total, those patients made 61 ED visits and contributed 276 hours of ED nosocomial exposure time generating 31 secondary cases. This represents 60.8% of all secondary cases attributed to the study cohort; 15 non-attendees generated 20 secondary cases. Moreover, ED attendees were more likely than non-ED attendees to be clustered (27% vs 21%, p = 0.037)–see Fig 2. Among the total of 845 ED visits, there were 18 visits that overlapped in time and place; i.e. nine pairs of unique patients attended the same ED on the same day. One of these pairs shared the same DNA fingerprint and we cannot exclude the possibility that transmission from one to the other took place at that time.

Compared to ED-attendees with single visits, ED-attendees with multiple visits were more likely to be Canadian-born (32.1% vs 17.2%), have a rural residence (43.1% vs, 21.7%), and have a social risk factor (15.5% vs. 6.4%), and were more likely to die before completion of treatment (9.3% vs. 3.2%).

The cumulative nosocomial exposure time generated by ED-attendees by month and smear-status at diagnosis is shown in Fig 3. There were 3812 total nosocomial exposure hours, 2999 (78.7%) in the three months immediately preceding diagnosis. More than half of the total and smear-positive nosocomial exposure hours occurred in the month immediately preceding the date of diagnosis. Assuming that the ED-attendees' first ED visit was their first encounter with the health system, the median number of days from first visit to diagnosis was 27 days (IQR: 7, 180 days).

Of the 292 ED-attendees who had a final visit within 30 days of their PTB diagnosis, 185 (63.4%) had a discharge diagnosis of either PTB or pulmonary 'other', thereby defined as being 'symptomatic' (Fig 4). When these symptomatic PTB patients had an epidemiologic risk

**Table 1. Demographic and clinical characteristics associated with attendance at an ED by subsequently diagnosed pulmonary TB patients.**

|  | Total | ED Attendee (%) | Non-Attendee (%) | p-value |
|---|---|---|---|---|
| **No. Assessed** | 711 | 378 (100) | 333 (100) | — |
| **Age (mean [SD])** | 711 | 48.6 [20.9] | 47.5 [19.2] | 0.47 |
| **Sex** |  |  |  |  |
| Male | 395 | 221 (58.4) | 174 (52.2) | 0.09 |
| Female | 316 | 157 (41.6) | 159 (47.8) |  |
| **Population Group** |  |  |  |  |
| Canadian-Born | 130 | 94 (24.8) | 36 (10.8) | <**0.01** |
| Foreign-Born | 581 | 284 (75.2) | 297 (89.2) |  |
| **Place of Residence*** |  |  |  |  |
| Urban | 522 | 255 (67.4) | 267 (80.1) | <**0.01** |
| Rural | 189 | 123 (32.6) | 66 (19.9) |  |
| **Disease Site** |  |  |  |  |
| PTB Alone | 592 | 298 (78.8) | 294 (88.2) | <**0.01** |
| PTB+EPTB | 119 | 80 (21.2) | 39 (11.8) |  |
| **Smear Status** |  |  |  |  |
| Negative | 357 | 155 (41.1) | 202 (60.7) | <**0.01** |
| Positive | 354 | 223 (58.9) | 131 (39.3) |  |
| **Drug Resistance** |  |  |  |  |
| No | 637 | 341 (90.3) | 296 (88.9) | 0.56 |
| Yes | 74 | 37 (9.7) | 37 (11.1) |  |
| **HIV Status** |  |  |  |  |
| Negative | 663 | 350 (92.5) | 313 (93.9) |  |
| Positive | 35 | 22 (5.8) | 13 (3.9) | 0.07 |
| Unknown[†] | 13 | 6 (1.7) | 7 (2.2) |  |
| **Diabetes[‡]** |  |  |  |  |
| No | 595 | 312 (82.6) | 283 (85.0) | 0.37 |
| Yes | 116 | 66 (17.4) | 50 (15.0) |  |
| **Social Risks** |  |  |  |  |
| No | 656 | 336 (88.9) | 320 (96.1) | <**0.01** |
| Yes | 55 | 42 (11.1) | 13 (3.9) |  |
| **Secondary Cases** |  |  |  |  |
| No | 674 | 356 (94.2) | 318 (95.5) | 0.43 |
| Yes | 37 | 22 (5.8) | 15 (4.5) |  |
| **Outcome** |  |  |  |  |
| Survived | 674 | 350 (92.5) | 324 (97.2) |  |
| Deceased | 28 | 24 (6.3) | 4 (1.2) | <**0.01** |
| Unknown[¶] | 9 | 4 (1.2) | 5 (1.6) |  |

Abbreviations: ED emergency department; TB tuberculosis; SD standard deviation; PTB pulmonary tuberculosis; EPTB extrapulmonary tuberculosis; HIV human immunodeficiency virus.

* Place of residence was the patient's 'usual place of residence' or 'where they lived most of the time' at diagnosis.

[†] Alberta implemented opt-out HIV testing of all tuberculosis patients in 2003. Of the 13 patients whose HIV status was unknown, 6 people refused testing, 4 people were deceased and 3 people were not offered testing for reasons that are unknown. Of the 22 HIV positive ED attendees, 16 (72.7%) were previously positive, and six were determined to be HIV positive at the time of their PTB diagnosis.

[‡] Of the 66 ED attendees who were diabetic, 60 (90.9%) were previously diagnosed and six were determined to be diabetic at the time of their PTB diagnosis.

[¶] All of the 9 patients whose treatment outcomes were unknown transferred outside of Canada prior to treatment completion.

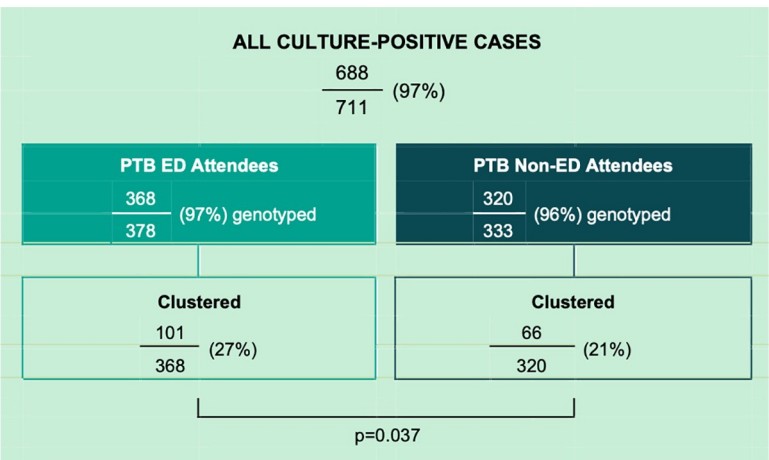

**Fig 2. Genotypic clustering among *M. tuberculosis* isolates from ED attendees versus non-attendees, 2010–2016.**

factor, and had undergone a CXR that was determined to be typical, PTB was either diagnosed or suspected in 76.7% (69/90); 80% (60/75) in patients ultimately diagnosed with smear-positive PTB, 60% (9/15) in patients ultimately diagnosed with smear-negative PTB.

## Discussion

In the Canadian Province of Alberta, the ED was a frequent place of contact for patients with undiagnosed PTB. While more than 50% of adult patients with PTB accessed the ED in the months immediately preceding their PTB diagnosis, making 845 visits, a discharge diagnosis of PTB occurred at only one in 10 of those visits. Yet, these ED visits by undiagnosed patients with PTB constituted <0.007% of the total of 11,182,876 ED visits made by adults in the Province during the study period [16, 17]. This underscores the challenge of identifying a relatively rare disease in a high throughput setting. Although 431 of the 845 ED visits (51%) occurred in just five EDs, the most visits that any one of these EDs experienced was 110 –or, an average of 16 visits per year. Nevertheless, the potential individual and public health consequences of missing a diagnosis of PTB during these visits was substantial.

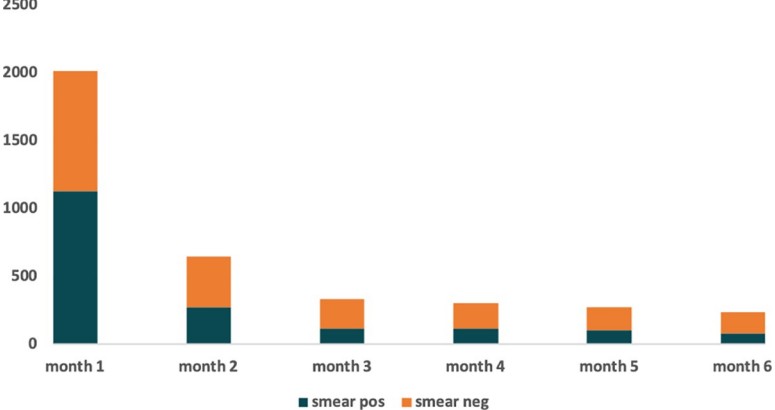

**Fig 3. Nosocomial exposure hours generated during ED visits by discharge pulmonary 'other' or non-pulmonary according to smear-status at eventual diagnosis.**

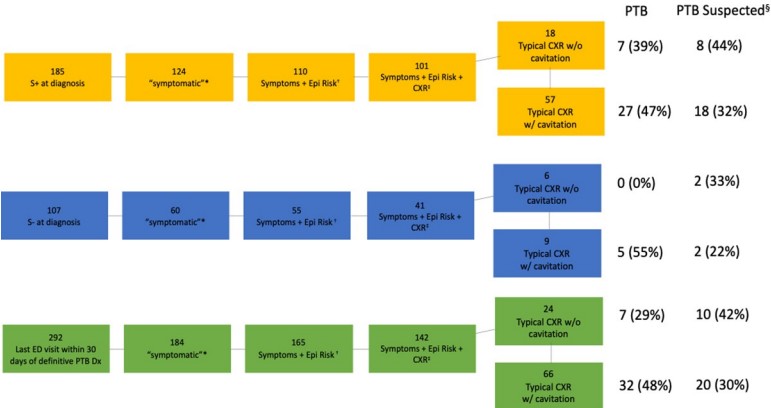

**Fig 4. Actual and potential yield of the combination of PTB diagnostic triggers in the ED, 30 days or less prior to the definitive diagnosis date.** Abbreviations: PTB pulmonary TB; ED emergency department; S+ smear-positive; S-smear-negative; Epi epidemiologic; CXR chest radiograph. *ED attendees whose final visit within the last 30 days and whose discharge diagnosis from the ED was either PTB or pulmonary 'other'. †See Box 1 for list of epidemiologic risks. ‡ 22 S+ and 23 S- ED-attendees did not have a chest radiograph. §"PTB suspected" = those who did not receive a most responsible discharge diagnosis of PTB, but in whom sputum/bronch specimen was submitted for acid-fast bacilli smear and culture from the ED visit.

If it is assumed that ED visits were the first encounter with the healthcare system, then ED-attendees had a median health system delay of 27 (IQR: 7, 180) days and, as a group, were more likely to die a TB-related death than non ED attendees (5.9% vs 1.2%). The finding that ED attendees were more likely to be smear-positive, to have co-existent extrapulmonary disease, and to have social risk factors, suggests that they might have had more severe disease or been more debilitated. ED attendees generated a total of 3812 hours of ED nosocomial exposure time, most of it in the three months immediately preceding their date of diagnosis when transmission risk is greatest [18], and generated 60.8% of the 51 secondary cases attributable to the entire study cohort. In addition, the likelihood of ED attendees belonging to a genotypically linked cluster (TB transmission chain) was greater than that of non-ED attendees.

The ED is uniquely well positioned to reduce delay in diagnosis and the negative individual and public health consequences of that delay [19] in addition to realizing cost-savings for the health care system. In a previous study, we showed ED-attendees place a much greater demand than non-attendees on health care services overall, independent of the ED–e.g. extra physician visits and hospitalizations that increase costs to the province [5]. That said, leveraging this position in low-incidence settings is enormously challenging. Similar to other studies, these visits were more likely to be for a respiratory complaint (discharge diagnosis of either PTB or pulmonary 'other') the closer they were to the diagnosis date [8]. Conversely, the majority of all visits were for non-pulmonary complaints (non-pulmonary discharge diagnoses) [8].

For purposes of infection control, suspicion of PTB would ideally occur at triage by implementation of a tool for screening and implementation of respiratory transmission precautions. With this in mind, and accepting that the three major triggers to suspicion of PTB are a combination of: i) symptoms, ii) epidemiologic risk, and iii) typical chest radiographic findings, only the first two are available at triage [9, 20–23].

Key symptoms are either respiratory: cough, especially if newly acquired and ≥ 2 weeks in duration, hemoptysis, and chest pain, or constitutional: fever, weight loss, and night sweats, or both. Epidemiologic risks are multifactorial and generally poorly assessed (see Box 1), especially in a busy ED. The sensitivity of such a screen in Alberta would be high for smear-positive PTB (95% of such cases have been shown to be symptomatic, and 95% have been shown to

Box 1. Epidemiological risk of exposure to or infection with *Mycobacterium tuberculosis*

- A history of tuberculosis and whether it was adequately treated

- A history of overt PTB contact and whether it was adequately assessed (e.g. tuberculin skin test, interferon-γ release assay) and treated

- Migration from a high TB incidence country*

- Travel to or work within a high TB incidence country

- Indigenous ancestry and/or residence/origin in high incidence Indigenous community

- Occupational history, e.g. healthcare worker, laboratory technician

- One or more social risk factors including: homelessness within 12 months of diagnosis, incarceration at diagnosis, and or record of substance misuse disorders where non-overt exposures might occur

*High TB incidence country was defined according to the 7th Edition of the Canadian TB Standards [23]

have an epidemiologic risk factor) but low for smear-negative PTB (only 63% have been shown to be symptomatic; 97% have been shown to have an epidemiologic risk factor) [22]. When a single ED in Los Angeles with approximately 108 PTB patient visits per year (both smear-positive and smear-negative) implemented such a screen, the sensitivity was only 63% [4]. Given these considerations and knowing that a triage screen would be difficult to administer in many patients–for example, those who are severely ill, cognitively impaired or unresponsive—this is an unrealistic option even in the five high PTB throughput EDs in Alberta.

Those patients whose ED visits occurred within 30 days of starting treatment suggest that among many, the triggers for a diagnosis were often present. A typical CXR in a symptomatic patient with an epidemiological risk, however, was not sufficient to guarantee a discharge diagnosis of PTB. Given the extraordinarily low prevalence of PTB in Alberta EDs, education of frontline providers in emergency medicine is not likely to improve diagnosis. This is especially so because there are few chances to reinforce recognition of a clinical picture of PTB. That said, treating the combination of these triggers as a 'syndrome' for PTB identifiable in public health records may leverage the power of a single electronic medical record (EMR) to support the process. In other words, an algorithm integrating clinical, epidemiologic, and radiologic features has the potential to identify virtually all smear positive-cases with a typical chest radiograph–the subset of patients with PTB who are by far the most likely to transmit the organism [14] from the vast administrative data within the EMR. This needs to be explored in future research.

Strengths of our study include its long duration and completeness of *M. tuberculosis* public health and genotyping data, allowing us to capture the ED experience of patients with PTB province-wide over many years, linkage to valid, population-based ED presentation data for the province, and detailed primary data on risk factors including Indigenous identity which is systematically recorded in the Provincial TB registry. Weaknesses include the retrospective

study design, and limited amount of visit-specific clinical (e.g. symptoms), investigative and treatment information in the records. The method we used to ascribe secondary cases to presumptive source cases may have underestimated transmission events occurring outside of the defined spatial-temporal limits, including any that may have happened in hospital. Herein we reported potential nosocomial exposure hours by smear-status; it is likely that in the near future, a the cycle threshold output of the Gene Xpert test will form a new proxy for infectiousness that replaces AFB-smear. Finally, when calculating the nosocomial exposure hours we assumed that PTB visits were associated with appropriate respiratory isolation precautions thereby limiting the contributions by these patients, but that may not have always been the case suggesting these hours are underestimated.

This study identifies the ED as a common location of care for patients with PTB, the most infectious form of TB. These data suggest the diagnosis is missed in 90% of cases therein; these missed diagnoses can lead to dangerous exposures for health care workers, additional spread of infections to close contacts, costs to the health care system, delayed diagnosis and poor outcomes. Efforts to identify interventions to improve these statistics are needed, especially in urban centres.

## Supporting information

**S1 Text.**
(TXT)

## Acknowledgments

The authors are very grateful to Natalie Runham, RN and research coordinator in the Emergency Medicine Research Group (EMeRG; Department of Emergency Medicine, University of Alberta), the staff of the Tuberculosis Program Evaluation and Research Unit, Data Analytics of Alberta Health Services, and Health Records for assistance in assembling the data presented in this study.

## Author Contributions

**Conceptualization:** Courtney Heffernan, Richard Long.

**Data curation:** Courtney Heffernan, Mary-Lou Egedahl.

**Formal analysis:** Courtney Heffernan, Catherine Paulsen, Leyla Asadi, James Barrie.

**Methodology:** Courtney Heffernan, Brian H. Rowe, Richard Long.

**Resources:** Mary-Lou Egedahl, Brian H. Rowe, James Barrie, Richard Long.

**Software:** Leyla Asadi.

**Validation:** Catherine Paulsen, Brian H. Rowe, James Barrie, Richard Long.

**Writing – original draft:** Courtney Heffernan.

**Writing – review & editing:** Courtney Heffernan, Catherine Paulsen, Leyla Asadi, Mary-Lou Egedahl, Brian H. Rowe, James Barrie, Richard Long.

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
