## [Decision Letter · Decision Letter 0]

15 Feb 2021

PONE-D-21-00447

INDIVIDUAL AND PUBLIC HEALTH CONSEQUENCES ASSOCIATED WITH A MISSED DIAGNOSIS OF PULMONARY TUBERCULOSIS IN THE EMERGENCY DEPARTMENT: A RETROSPECTIVE COHORT STUDY

PLOS ONE

Dear Dr. Heffernan,

Thank you for submitting your manuscript to PLOS ONE. After careful consideration, we feel that it has merit but does not fully meet PLOS ONE’s publication criteria as it currently stands. Therefore, we invite you to submit a revised version of the manuscript that addresses the points raised during the review process.

We look forward to receiving your revised manuscript.

Kind regards,

Seyed Ehtesham Hasnain

Academic Editor

PLOS ONE

Additional Editor Comments:

Minor Revision

Journal Requirements:

2. Please note that all PLOS journals ask authors to adhere to our policies for sharing of data and materials: https://journals.plos.org/plosone/s/data-availability. According to PLOS ONE’s Data Availability policy, we require that the minimal dataset underlying results reported in the submission must be made immediately and freely available at the time of publication. As such, please remove any instances of 'unpublished data' or 'data not shown' in your manuscript and replace these with either the relevant data (in the form of additional figures, tables or descriptive text, as appropriate), a citation to where the data can be found, or remove altogether any statements supported by data not presented in the manuscript.

Reviewers' comments:

Reviewer's Responses to Questions

**Comments to the Author**

1. Is the manuscript technically sound, and do the data support the conclusions?

Reviewer #1: Yes

Reviewer #2: Yes

2. Has the statistical analysis been performed appropriately and rigorously? 

Reviewer #1: Yes

Reviewer #2: I Don't Know

3. Have the authors made all data underlying the findings in their manuscript fully available?

Reviewer #1: Yes

Reviewer #2: Yes

4. Is the manuscript presented in an intelligible fashion and written in standard English?

Reviewer #1: Yes

Reviewer #2: Yes

5. Review Comments to the Author

Reviewer #1: Comments:

Study conducted by Heffernan C et al has been designed very nicely to find out the missed diagnosed TB patients at the level of emergency departments. This will create health hazardous and dangerous exposures to the health care workers as well as the house hold contacts, especially in pediatrics groups. In case of previously treated TB patients, this may developed drug-resistant TB among the common population.

I am happy to accept this article with single correction.

Method:

• L. no. 102 add reference.

Reviewer #2: The specificity and sensitivity of smear test for diagnosis of TB.

Since it was a retrospective cohort study therefore the standard mode is acceptable. But the authors view on advances in diagnosis of TB such as gene expert.

6. PLOS authors have the option to publish the peer review history of their article (what does this mean?). If published, this will include your full peer review and any attached files.

Reviewer #1: No

Reviewer #2: No

---

## [Author Response · Author response to Decision Letter 0]

22 Feb 2021

We thank the editorial board, and the reviewers for providing us with feedback to make a resubmission of this work possible. We have responded to the issues raised by the Editorial board, and by the reviewers, as below.

We have reformatted our manuscript and our files according to the guidelines provided.

2. Please note that all PLOS journals ask authors to adhere to our policies for sharing of data and materials: https://journals.plos.org/plosone/s/data-availability. According to PLOS ONE’s Data Availability policy, we require that the minimal dataset underlying results reported in the submission must be made immediately and freely available at the time of publication. As such, please remove any instances of 'unpublished data' or 'data not shown' in your manuscript and replace these with either the relevant data (in the form of additional figures, tables or descriptive text, as appropriate), a citation to where the data can be found, or remove altogether any statements supported by data not presented in the manuscript.

We have removed the statement supported by the text, “data not shown” as it was not directly relevant to our findings.

Reviewer #1: Comments:

Study conducted by Heffernan C et al has been designed very nicely to find out the missed diagnosed TB patients at the level of emergency departments. This will create health hazardous and dangerous exposures to the health care workers as well as the house hold contacts, especially in pediatrics groups. In case of previously treated TB patients, this may developed drug-resistant TB among the common population.

I am happy to accept this article with single correction.

Method:

• L. no. 102 add reference.

Thank you for the careful review – the text ‘ref’ was included in the text in error, so we have deleted this text rather than adding an additional reference. 

Reviewer #2: The specificity and sensitivity of smear test for diagnosis of TB.

Since it was a retrospective cohort study therefore the standard mode is acceptable. But the authors view on advances in diagnosis of TB such as gene expert.

Thank you for this comment; we have added a sentence (see lines: 345-348) about the future potential of Gene Xpert in supporting the diagnosis of pulmonary TB.

---

## [Editor Report · Decision Letter 1]

1 Mar 2021

INDIVIDUAL AND PUBLIC HEALTH CONSEQUENCES ASSOCIATED WITH A MISSED DIAGNOSIS OF PULMONARY TUBERCULOSIS IN THE EMERGENCY DEPARTMENT: A RETROSPECTIVE COHORT STUDY

PONE-D-21-00447R1

Dear Dr. Heffernan,

We’re pleased to inform you that your manuscript has been judged scientifically suitable for publication and will be formally accepted for publication once it meets all outstanding technical requirements.

Kind regards,

Seyed Ehtesham Hasnain

Academic Editor

PLOS ONE

Additional Editor Comments (optional):

I have gone through the revised manuscript and also the Authors response to the comments of the reviewers. The manuscript was sent for Minor revision and Authors have modified the manuscript keeping in mind the comments of the Reviewers. As suggested by the Reviewers, Authors have added a sentence in line number 345-348 about the future potential of Gene Xpert in supporting the diagnosis of pulmonary TB. In my view, the authors have satisfactorily addressed all the comments made by the reviewers and have revised the manuscript accordingly. I recommend this manuscript for publication.
---

## [Editor Report · Acceptance letter]

12 Mar 2021

PONE-D-21-00447R1 

Individual and Public Health Consequences Associated with A Missed Diagnosis of Pulmonary Tuberculosis in The Emergency Department: A Retrospective Cohort Study 

Dear Dr. Heffernan:

I'm pleased to inform you that your manuscript has been deemed suitable for publication in PLOS ONE. Congratulations! Your manuscript is now with our production department. 

Kind regards, 

on behalf of

Dr. Seyed Ehtesham Hasnain 

Academic Editor

PLOS ONE